# A Natural Quinazoline Derivative from Marine Sponge *Hyrtios erectus* Induces Apoptosis of Breast Cancer Cells via ROS Production and Intrinsic or Extrinsic Apoptosis Pathways

**DOI:** 10.3390/md17120658

**Published:** 2019-11-23

**Authors:** Arun Kumar De, Ramachandran Muthiyan, Samiran Mondal, Nilkamal Mahanta, Debasis Bhattacharya, Perumal Ponraj, Kangayan Muniswamy, Anandamoy Kundu, Madhu Sudhan Kundu, Jai Sunder, Dhanasekar Karunakaran, Asit Kumar Bera, Sibnarayan Dam Roy, Dhruba Malakar

**Affiliations:** 1ICAR-Central Island Agricultural Research Institute, Port Blair 744101, Andaman and Nicobar Islands, India; debasis63@rediffmail.com (D.B.); perumalponraj@gmail.com (P.P.); swamy02_vet@yahoo.co.in (K.M.); drakundu61@gmail.com (A.K.); mkundu47@rediffmail.com (M.S.K.); jaisunder@rediffmail.com (J.S.); 2Bioinformatics Centre, ICAR-Central Island Agricultural Research Institute, Port Blair 744101, Andaman and Nicobar Islands, India; insilicobrain@gmail.com; 3Department of Veterinary Pathology, West Bengal University of Animal and Fisheries Sciences, Kolkata 700037, West Bengal, India; vetsamiran@gmail.com; 4Department of Chemistry and Institute for Genomic Biology, University of Illinois, Urbana Champaign, IL 61801, USA; nilkamal1984@gmail.com; 5Department of Chemistry, Indian Institute of Technology, Dharwad, WALMI campus (near High Court), Bangaluru 580011, Karnataka, India; 6Division of Social Science, ICAR-Central Island Agricultural Research Institute, Port Blair 744101, Andaman and Nicobar Islands, India; karunakaran_tn@rediffmail.com; 7Reservoir and Wetland Fisheries Division, ICAR-Central Inland Fishery Research Institute, Barrackpore, Kolkata 700120, West Bengal, India; asitmed2000@yahoo.com; 8Division of Fisheries Sciences, ICAR-Central Island Agricultural Research Institute, Port Blair 744101, Andaman and Nicobar Islands, India; sibnarayan@gmail.com; 9Animal Biotechnology Centre, National Dairy Research Institute, Karnal 132001, Haryana, India

**Keywords:** marine sponge, quinazoline derivative, apoptosis, breast cancer, ROS production

## Abstract

Here, we report the therapeutic potential of a natural quinazoline derivative (2-chloro-6-phenyl-8H-quinazolino[4,3-b]quinazolin-8-one) isolated from marine sponge *Hyrtios erectus* against human breast cancer. The cytotoxicity of the compound was investigated on a human breast carcinoma cell line (MCF-7). Antiproliferative activity of the compound was estimated by 3-(4,5-Dimethylthiazol-2-yl)-2,5-diphenyltetrazolium bromide (MTT) assay. MTT assay showed significant inhibition of MCF-7 cells viability with the IC50 value of 13.04 ± 1.03 µg/mL after 48 h. The compound induced down-regulation of anti-apoptotic Bcl-2 protein and increase in the pro-apoptotic Bax/Bcl-2 ratio in MCF-7 cells. The compound activated the expression of Caspases-9 and stimulated downstream signal transducer Caspase-7. In addition, Caspase-8 showed remarkable up-regulation in MCF-7 cells treated with the compound. Moreover, the compound was found to promote oxidative stress in MCF-7 cells that led to cell death. In conclusion, the compound could induce apoptosis of breast carcinoma cells via a mechanism that involves ROS production and either extrinsic or intrinsic apoptosis pathways. The systemic toxic potential of the compound was evaluated in an in vivo mouse model, and it was found non-toxic to the major organs.

## 1. Introduction

Breast cancer is a major cause of death in woman globally, with one-third of the women with breast cancer developing metastases and ultimately dying of the disease [1,2]. Every year, thousands of women are diagnosed with breast cancer globally [3], and its incidence is on a rising trend [4,5]. In 2019, an estimated 606,880 people will die from cancer, and 62,930 new cases of female breast carcinoma have expected to occur in United States [6]. Although advances in diagnosis and treatment have improved the survival rates in breast cancer patients, a very successful therapy for breast cancer has yet to be developed. Therefore, it is very important to explore novel sources for the development of an effective drug against breast cancer.

The marine ecosystem acts as a great repository of compounds with huge therapeutic potential. Bioactive compounds with different therapeutic applications, including cancer treatment, have been isolated from marine resources [7,8]. Marine sponges have been found to be a potent storehouse of chemicals with varied therapeutic potential. Marine sponges lead a sedentary lifestyle and protect themselves from predators by synthesizing different chemical compounds [9]. Symbiotic microorganisms that inhabit inside sponges are greatly responsible for the production of these compounds [10,11,12]. Due to the huge potential of marine-derived drugs, especially in the treatment of cancer, the investigation to find novel bioactive compounds from the marine environment is rapidly growing [13]. As of 2012, more than 15,000 marine chemical compounds have been isolated and tested [14]. Bioactive compounds with antimicrobial, anti-inflammatory, and antiviral applications have been isolated from the sponges of the genus *Hyrtios* [15,16].

Quinazoline is a very predominant scaffold in many natural and synthetic bioactive compounds [17]. Therefore, research to discover novel quinazoline compounds effective in cancer treatment has been intensified [18]. Various types of pharmacological activities of quinazoline derivatives, like anti-cancer [19], anti-oxidant [20], anti-viral [21], anti-convulsant [22], anti-inflammatory [23], and anti-tubercular [24] activities, have been reported. In the present study, we evaluated the anticancer potential of a natural quinazoline derivative (Compound **A**) against a breast carcinoma cell line (MCF-7). The mechanism of action of the derivative was also investigated. An acute toxicity test using a mice model was done to assess the in vivo toxic potential of the compound.

## 2. Results

### 2.1. Compound ***A*** Inhibits the Growth of Human Breast Carcinoma Cells (MCF-7) In Vitro

An MTT cytotoxicity assay was performed to find out the anti-proliferative effect of the compound **A** on MCF-7 cells. Exponentially growing cells were exposed to various concentrations of the compound for 24 h and 48 h. The results showed that the compound **A** inhibited the proliferation of the MCF-7 cells in a concentration-dependent manner (Figure 1A,B). The determined half maximal inhibitory concentration values (IC50) of compound **A** on MCF-7 cells were 22.67 ± 1.53 µg/mL and 13.04 ± 1.03 µg/mL for 24 h and 48 h, respectively (Table 1). On the other hand, the IC50 values of the compound on a non-tumorigenic epithelial cell line (MCF-10A) were 102.11 ± 1.89 µg/mL and 51.25 ± 1.42 µg/mL for 24 h and 48 h, respectively (Figure 1C,D, Table 1), indicating that the compound **A** is relatively less cytotoxic toward non-tumorigenic epithelial cells as compared to breast carcinoma cells. The role of oxidative stress in compound **A** induced apoptosis was investigated by pre-treatment of the cells with antioxidant ascorbic acid prior to treatment with the compound. Pre-treatment of the MCF-7 cells with ascorbic acid increased the viability of MCF-7 cells treated with the compound **A** in a dose-dependent manner (Figure 1A,B). Cyclophosphamide was used as a standard anticancer drug. The IC50 values of cyclophosphamide on MCF-7 cells were 15.11 ± 1.16 µg/mL and 8.11 ± 0.84 µg/mL for 24 h and 48 h respectively. The IC50 values on MCF-10A cells were 59.23 ± 1.68 µg/mL and 26.22 ± 1.07 µg/mL for 24 h and 48 h respectively (Table 1). We also investigated the effect of the compound **A** on the colony forming potential of the MCF-7 cells, and it was found that the compound reduced colony forming potential of breast carcinoma cells in a concentration-dependent manner (Figure 1E). Morphological changes were observed by phase contrast microscopy (Figure 2). At 24 h after treatment, a decrease in total cell number and the increase in floating cells were observed. The cytotoxic potential of compound **A** on two other mammary adenocarcinoma cell lines (MDA-MB-231 and MDA-MB-415) were also investigated, and it was found that the compound inhibited the growth of both the cell lines in a concentration-dependent manner (Appendix A). Collectively, these results indicated that compound **A** has a selective cytotoxic activity against breast carcinoma cells.

### 2.2. Lactate Dehydrogenase (LDH) Cytotoxicity Test

Leakage of intracellular lactate dehydrogenase (LDH) in the medium has been considered as an indicator of the disruption of the plasma membrane [25]. When the plasma membrane is damaged due to apoptosis, necrosis, or due to any other damage, LDH is rapidly released into the cell culture supernatant [26]. The effect of the compound on the release of LDH was assessed after incubation of the compound for 48 h. The results showed that the compound at a concentration as low as 25 µg/mL induced a significant increase in LDH release in treated cells compared to control cells (Figure 3), indicating a loss of membrane integrity of the treated cancer cells.

### 2.3. The Compound ***A*** Induces Apoptosis of Cancerous MCF-7 Cells

Reduction in MCF-7 cell viability following treatment with the quinazoline derivative suggested the possibility of cell death. To determine the cytotoxic mechanism of compound **A**, the effect of the compound on apoptosis of the MCF-7 cells was investigated. The results depicted that the compound induced apoptosis of the MCF-7 cells in a concentration-dependent manner. MCF-7 cells cultured in the absence of the compound showed 6.41% apoptotic cells, which increased to 18.45, 22.94, and 38.67% following the addition of 15, 25, and 50 µg/mL of compound **A** for 24 h treatment (Figure 4A–D,G). To confirm the role of oxidative stress in the compound **A** induced apoptosis, the effect of pre-treatment of the cells with antioxidant ascorbic acid prior to treatment with the compound was analyzed. It was found that pre-treatment of the MCF-7 cells with ascorbic acid reduced the percentage of apoptotic cells, confirming the involvement of oxidative stress in compound **A**–mediated apoptosis (Figure 4E–G). Further, the apoptotic potential of compound **A** on two other mammary adenocarcinoma cell lines (MDA-MB-231 and MDA-MB-415) was investigated, and it was found that the compound induced apoptosis of both the cell lines (Appendix A), indicating its apoptotic potential against breast carcinoma cell lines. On the other hand, in MCF-10A cells, no significant difference of apoptotic cell percentage between compound **A**–treated cells and control was found (Appendix A), indicating selective cytotoxicity against breast carcinoma cells.

### 2.4. Generation of Reactive Oxygen Species (ROS)

As reactive oxygen species (ROS) play an important role in apoptosis [27], the level of ROS was evaluated in the treated MCF-7 cells with different concentrations of the quinazoline derivative. Cells treated with 400 µg/mL of tert-Butyl hydroperoxide (TBHP) were taken as positive control [28]. The result is presented in Figure 5. The incubation of MCF-7 cells with the compound generated significantly higher ROS at a concentration of 25 µg/mL onward, as compared control cells. Treatment of MCF-7 cells with 0.1 mM ascorbic acid for 6 h before treatment with the compound attenuated the formation of intracellular ROS. On the other hand, in MCF-10A cells, no significant difference in ROS production between treated cells and control cell was observed (Appendix A).

### 2.5. Effects of Compound ***A*** on the Expression of Apoptotic-Related Genes in MCF-7 Cells

To gain a further insight into the mechanism underlying compound **A**–induced apoptosis, the expression profiles of 12 genes involved in the apoptotic pathway were evaluated by real-time PCR. As shown in Figure 6, the expression levels of pro-apoptotic genes *BAX, BID, TR53, CDKN1A (p21), CASP2, CASP7, CASP8, CASP9*, and *PARP1* were up-regulated, while the expression levels of anti-apoptotic genes *BCL2, BCLXL,* and *MCL1* were down-regulated following treatment with the compound. Significant differences in the expressions of *BAX, TP53, CASP7, CASP9*, and *BCLXL* were observed at concentration as low as 15 µg/mL, while the difference in the expression level of the rest of the genes required a little higher concentration of the compound (25 µg/mL). The ratio of BAX to BCL2 gene in control and compound-treated MCF-7 cells is presented in Figure 7. Treatment of the compound at a concentration of 25 µg/mL induced a significant increase in the BAX to BCL2 gene ratio in MCF-7 cell line. Overall, the results of the study indicated that the compound induces up-regulation of pro-apoptotic genes and down-regulation of anti-apoptotic genes in breast carcinoma MCF-7 cell line. The effect of the standard anticancer drug cyclophosphamide on the expression of apoptosis related genes in presented in Appendix A. It was observed that the expression levels of *BAX, TR53, CDKN1A (p21), CASP8*, and *PARP1* were up-regulated, while expression level of *BCL2* and *MCL1* were down-regulated following treatment with cyclophosphamide. The expression level of *CASP9* was unchanged.

### 2.6. Effects of the Compound ***A*** on the Expression of Apoptotic-Related Proteins in MCF-7 Cells

To further support the gene expression results, Western blot analysis was performed to determine the effect of the compound on the expression of five proteins involved in apoptotic pathway (Bcl-2, Bax, Caspase 7, Caspase 9, and PARP1). Figure 8 depicted that after treatment with the compound, the expression levels of Bax, Caspase 7, and Caspase 9 were up-regulated, but Bcl-2 was down-regulated. Treatment of the MCF-7 cells induced cleavage of Poly (ADP-ribose) polymerase (PARP) (Figure 8A), which is considered as a hallmark of apoptosis. The ratio of Bax to Bcl-2 protein in compound-treated MCF-7 cells was higher compared to the control (Figure 8F). 

### 2.7. Effect of the Compound ***A*** on Caspase 8 Activity

A Caspase 8 activity assay was done to investigate the effect of the compound **A** on Caspase 8 activity. It was observed that the compound induced the activity of Caspase 8 (Figure 9). A significant up-regulation of Caspase 8 activity was observed in MCF-7 cells treated with the compound either at 25 µg/mL and 50 µg/mL concentration as compared to control.

### 2.8. Acute Toxicity Study

Acute toxicity of the compound was tested in a mouse model. Mice were fed with the quinazoline derivative at 250 mg/kg, single dose, and observed for 14 days, after which histopathological changes in major organs were investigated. The treatment group showed hurdling for the first 1 h post dosing, then no abnormality in behavior was observed. There was no death during the experiment. Grossly, all organs of the treated group were same as like those of control group. In histological analysis, kidney showed mild focal desquamation of tubular epithelial cell in the distal convoluted tubule in the treatment group (Figure 10B). Glomerulus within the renal capsule was well defined, and the vascular pole was also present, although there was focal coiling of capillary tuft (Figure 10A,B). Proximal convoluted tubular epithelium was cuboidal, and these were also in continuity with the capsular membrane in both control and treatment group (Figure 10A,B). Histopathology of the liver of the treatment group (Figure 10D) showed very mild reversible changes of clear cell hepatocyte. The arrangement of the hepatic chord was unchanged. Central vein, bile duct epithelium, and sinusoids were normal as compared to the control group (Figure 10C,D). Very mild degenerative changes were present in hepatocyte, though focal in nature, in the treatment group (Figure 10D). There were no changes in spleen in comparison to control animal (Figure 10E,F). White pulp and red pulp were both normal. In the intestine, the anatomic pathology of payer’s patch, villi, lamina propia, layer muscular, and serosal layer were the same in comparison to control mice (Figure 10G,H). The pyramidal cells of the brain were normal basophilic in both control and treatment groups (Figure 10I,J). The cellularity and texture of white matter was normal (Figure 10I,J). Overall, no significant changes in the histology of the major organs were observed in mice fed with compound **A** (Figure 10). Serum total protein, albumin, globulin, and ALT and AST levels of control and treatment mice were estimated to assess liver function, and no significant difference between the control and treated groups was found in any of these parameters (Table 2). 

## 3. Discussion

In the present study, the cytotoxic potential of a sponge-derived quinazoline derivative (2-chloro-6-phenyl-8H-quinazolino[4,3-b]quinazolin-8-one) against a breast carcinoma cell line (MCF-7) was investigated, and it was observed that the compound induced apoptosis of MCF-7 cells. The MCF-7 cell line is being used as an in vitro model for breast cancer–related studies [30]. In this study, we found that the compound, dependent on concentration, inhibited MCF-7 cell growth and proliferation, indicating its cytotoxic effect. The compound was found to be cytotoxic against another two mammary adenocarcinoma cell lines (MDA-MB-231 and MDA-MB-415). On the other hand, the compound was less cytotoxic toward MCF-10A cells compared to MCF-7 cells (Table 1), indicating its selective cytotoxic activity toward cancer cells. MCF-10A is a non-transformed epithelial cell line originated from human mammary tissue [31]. The results indicated that the compound selectively inhibited the growth of the breast cancer cells, which is a desirable property of an ideal anticancer agent [32]. In this study, the quinazoline derivative was further investigated to deduce the mode of cell death, and it was found that the compound indicated apoptotic cell death via ROS production and intrinsic or extrinsic apoptotic pathways.

Generation of oxidative stress is considered to be one of the mediators of apoptosis [33]. ROS plays an important role in induction of apoptosis in several cancers; it interferes with mitochondrial membrane potential, leading to release of cytochrome c, which activates Caspase9 expression [27,34]. ROS stimulates the oxidation of mitochondrial membrane pores, leading to interference of mitochondrial membrane potential, which may initiate the release of cytochrome c [35]. The involvement of oxidative stress in the compound-induced apoptosis was confirmed as pre-treatment of the MCF-7 cells with ascorbic acid increased the viability of MCF-7 cells treated with the compound in a dose-dependent manner (Figure 1A,B). It was also found that pre-treatment of the MCF-7 cells with ascorbic acid reduced the percentage of apoptotic cells confirming the involvement of oxidative stress in compound **A** mediated apoptosis (Figure 4E–G). Induction of ROS formation in MCF-7 cells following the treatment of the compound has been recorded in the present study (Figure 5). Pre-treatment of MCF-7 cells with ascorbic acid attenuated the formation of intracellular ROS (Figure 5), indicating that oxidative stress played a role in the compound-induced apoptosis of MCF-7 cells. The mechanism of ROS generation by the quinazoline derivative merits further study. Moreover, an up-regulation of TP53 (p53) and p21 genes was observed in the compound-treated MCF-7 cells (Figure 6). Under cellular stress, p53 triggers an up-regulation of P21 gene [36,37].

Apoptosis is a normal biological process that maintains homeostasis in cells [38,39]. A group of cysteine proteases known as caspases plays the most vital roles in apoptosis signaling pathways. Two major pathways of apoptosis have been reported—a) the mitochondrial pathway and b) the extrinsic pathway [40]. In the mitochondrial pathway, cytochrome c is released from mitochondria and stimulates Caspase9, whereas extrinsic pathway leads to activation of Caspase8. Both the pathways lead to activation of downstream Caspase3, which subsequently stimulates the signal for apoptosis of cells [40]. MCF-7 cells have lost Caspase3 expression due to a 47 base-pair deletion in exon 3 of casp-3 gene [41]. It has been observed that Caspase7 can also lead to apoptosis induction in MCF-7 cells in the absence of Caspase3 [41,42], as Caspase7 is highly related to Caspase3 in terms of substrate specificity [43]. In the present study, it was found that the compound induced increased gene expression of Caspase 2, 7, 8, and 9 (Figure 6). In Western blot analysis, increased expression of Caspase 7 and 9 was detected in the compound treated breast cancer cells (Figure 8), indicating its involvement in both intrinsic and extrinsic pathway of apoptosis. Involvement of extrinsic pathway of apoptosis was further confirmed as exposure of compound **A** stimulated activity of Caspase 8 (Figure 9). Therefore, the most possible mechanism of compound **A**-induced apoptosis of breast carcinoma cells is via ROS generation and involvement of either intrinsic or extrinsic apoptotic pathways.

The BCL-2 family of proteins is reported to be major players in anti-apoptosis [44,45]. They are key regulatory factors of the mitochondrial pathway [46]. The over-expression of anti-apoptotic Bcl-2 and the reduced expression of pro-apoptotic Bax is common in many human cancers [47,48,49]. Up-regulation of Bax alone can trigger a cell to apoptosis [50]. In the present study, an increase in the ratio of both Bax to Bcl 2 gene and protein in MCF-7 cells has been observed following exposure to the compound (Figure 7 and Figure 8). The expression of Bax is an early event that sensitizes the cell to undergo apoptosis. An increase in the expression of Bax, and the decrease in the expression of Bcl-2 were observed in MCF-7 cells following treatment with the compound (Figure 6 and Figure 8). Bax can cause mitochondria to release cytochrome c [51]. Bax induces mitochondria to open pores in the outer membrane, allowing the release of cytochrome c. Significantly higher levels of Caspase 2, 7, 8, and 9 were observed in MCF-7 cells following exposure of the compound which indicated that both intrinsic and extrinsic apoptosis pathways were associated with the apoptosis of MCF-7 cells. Poly (ADP-ribose) polymerase (PARP-1) plays a vital role in caspase-mediated cell death [52]. One of the hallmarks of apoptosis is the cleavage of PARP-1 by caspases [53,54]. In the present study, treatment of the MCF-7 cells induced cleavage of PARP-1 (Figure 8). Taken together, the most likely sequence of events responsible for the compound induced apoptosis in MCF-7 cells is presented in Figure 11.

Marine sponges of the genus *Hyrtios* have been well known as rich reservoir of unique bioactive products [55]. Several bioactive compounds with different therapeutic potential have already been isolated from this sponge genus [15,56]. Therefore, to further explore the potentiality of the sponge *Hyrtios erectus*, we are reporting the therapeutic potential of the quinazoline derivative (2-chloro-6-phenyl-8H-quinazolino[4,3-b]quinazolin-8-one) in the treatment of breast carcinoma. A wide range of biochemical applications of quinazoline derivatives has been documented [57,58]. Several pharmacologically important compounds contain quinazoline nucleus as their basic framework [57]. Quinazoline derivatives have emerged as potential chemotherapeutic agents against solid tumors [59,60,61]. Several quinazoline derivatives like gefitinib, erlotinib, lapatinib, and afatinib have been approved by the FDA [62]. It was reported earlier that quinazoline compounds could selectively inhibit tumor cell proliferation; they were significantly less cytotoxic in normal cells [63,64]. The results of our present study also showed that compound **A** is relatively more cytotoxic toward breast carcinoma cell lines as compared to non-tumorigenic epithelial cells, which indicates its potential as a therapeutic agent in breast cancer treatment. However, further in-depth in vivo studies on the compound are required to fully understand its anticancer property/potentiality.

## 4. Materials and Methods

### 4.1. Reagents

All chemicals used were purchased from Sigma-Aldrich (St. Louis, MO, USA), and cell culture plastics were from Nunc (Roskilde, Denmark), unless stated otherwise.

### 4.2. Sample Collection and Extraction of the Bioactive Compound

The sponge specimens were sampled from the Andaman sea of Andaman group of islands by SCUBA diving in the year 2016. The samples were immediately placed in methanol and brought to the laboratory and stored at −80 °C until further analysis. Extraction and characterization of the compound was done as previously reported [65].

### 4.3. Anticancer Activity of Quinazoline Derivative

#### 4.3.1. Culture of Cell Line

The human breast carcinoma cell lines (MCF-7, MDA-MB-231, and MDA-MB-415) and non-tumorigenic epithelial cells (MCF-10A) were obtained from National Centre for Cell Sciences (NCCS), Pune, India, and cultured in Dulbecco’s modified Eagle’s medium supplemented with 10% Fetal Bovine Serum (FBS), 2 mM glutamine, and 100 µg/mL penicillin-streptomycin. Cells were cultured in tissue culture flasks at 37 °C with 5% CO_2_ in the air. Exponentially growing cells were used for all the experiments.

#### 4.3.2. Cell Viability MTT (4,5-Dimethyl Thiazol-2-yl-2,5-Diphenyl Tetrazolium Bromide) Assay

MCF-7 and MCF-10A cells at a concentration of 1 × 10^6^ cells/well were cultured in a 96-well culture plate in DMEM medium for 24 h for cell attachment. Then different concentrations of the compound (0–500 µg/mL) were added to the medium and incubated at 37 °C and 5% CO_2_ tension in a humidified incubator for either 24 h or 48 h. Cells without the addition of the compound were treated as control. To find out the involvement of oxidative stress, cells were then pre-treated with 0.1 mM ascorbic acid [32] for 6 h before treatment with compound **A**. Then, the cells were treated with 100 µL of MTT solution (5 mg/mL) for 4 h for the formation of formazan crystals. After MTT treatment, the medium in each well was discarded, and DMSO (100 µL) was added to each well to solubilize the formazan crystals. Finally, the absorbance in each well was recorded at a wavelength of 540 nm using a microplate reader (SpectraMax Plus, Molecular Devices, San Jose, CA, USA) [66]. The cells were treated with cyclophosphamide, a standard anticancer drug that was used as a positive control.

#### 4.3.3. Lactate Dehydrogenase (LDH) Assay

Lactate dehydrogenase (LDH) release was measured by a commercial kit (PierceTM LDH Cytotoxicity Assay Kit, Thermo Scientific, Rockford, IL, USA) as per the manufacturer’s protocol. MCF-7 cells were incubated with medium containing different concentrations of the compound for 48 h. After that, the media of each well was taken out in a separate 96 well plate, mixed with the reaction mixture, and incubated for 30 min at room temperature. Then, the absorbance in each well was recorded at 490 nm using a microplate reader (SpectraMax Plus, Molecular Devices) [67].

#### 4.3.4. Apoptosis Assay

The FITC Annexin V/Dead Cell Apoptosis Kit (Invitrogen, Life Technologies, Camarillo, CA, USA) was used to measure the apoptotic potential of the compound on MCF-7 cells. After treatment of the cells with the compound for 48 h, the cells were stained with Annexin V-FITC and propidium iodide (Invitrogen, Life Technologies) as per manufacturer’s protocol. Then, the cells were kept at room temperature for 15 min and the fluorescence was measured by a flow cytometer (FAC Scan, Becton Dickinson). To find out the involvement of oxidative stress, cells were then pre-treated with 0.1 mM ascorbic acid for 6 h before treatment with compound **A**.

#### 4.3.5. Reactive Oxygen Species (ROS) Assay

An ROS assay was carried out to determine the effect of compound **A** on the production of ROS levels in treated MCF-7 cells. A fluorometric intracellular ROS kit (Sigma, USA) was used for the measurement of ROS activity. Briefly, 5 × 10^4^ cells per well were seeded into a 96-well plate and incubated overnight at a humidified incubator at 37 °C temperature and 5% CO_2_. The cells were then treated with different concentrations of the compound. Cells cultured without the compound were considered as control. After 24 h, 100 µL of ROS detection reagent master mix was added to each well and incubated for 1 h at 37 °C and 5% CO_2_ in an incubator. Cells treated with ROS inducer tert-Butyl hydroperoxide (TBHP at 400 µM) for 1 h under standard culture conditions were used as a positive control [28]. The fluorescence intensity was measured using a fluorescent plate reader at an excitation wavelength of 640 nm and an emission wavelength of 675 nm.

### 4.4. Determination of the Effects of Compound ***A*** on the Expression of the Apoptotic Pathway–Related Genes

#### 4.4.1. RNA Isolation

MCF-7 cells in logarithmic phage were treated with the compound (15, 25 and 50 µg/mL) for 24 h. Total cellular RNA was isolated from untreated control and treated cells using RNeasy Mini Kit (Qiagen, Valencia, CA, USA) as per the manufacturer’s protocol. Quality of the isolated RNA was assessed using a NanoDrop spectrophotometer (Thermo Scientific) and the RNA concentration was determined by Qubit^®^ RNA HS Assay Kit (Life Technologies, Camarillo, CA, USA) as per manufacturer’s protocol.

#### 4.4.2. Quantitative RT-PCR

Reverse transcription of RNA was done using a commercial kit named SuperScript III First-Strand Synthesis System for RT-PCR kit (Invitrogen, Life Technologies, Camarillo, CA, USA) as per manufacturer’s protocol. Relative expression of 12 genes (*BAX, BID, TP53, CDKN1A, CASP2, CASP7, CASP8, CASP9, PARP1, BCL2, BCLXL*, and *MCL1*) was quantified by real-time PCR using Power SYBR green PCR Master Mix (2X) (Applied Biosystems, Foster City, CA, USA) in a ABI 7900 Real-Time PCR system (Applied Biosystems). The housekeeping gene GAPDH was used as an endogenous control. Relative expression was determined using the 2-ΔΔCt method relative to the internal control. The information on the primers used is presented in Appendix A.

### 4.5. Determination of the Effects of Compound ***A*** on the Expression of Apoptotic Pathway–Related Proteins by Western Blotting

The expression of five proteins (Caspase7, Caspase 9, Bax, Bcl2, and PARP1) was analyzed by Western blotting. Cells from untreated control and treated cells (15, 25, and 50 µg/mL compound **A**) were lysed in ice-cold RIPA lysis buffer (50 mM Tris-HCl, 150 mM NaCl, 0.1% SDS (w/v), 0.5% sodium deoxycholate (w/v), 1% Triton X-100 (v/v), 1 mM PMSF, 10 µL/mL phosphatase inhibitor, and 10 µL/mL of protease inhibitor). The protein concentrations of cell lysate were determined by a colorimetric assay using Pierce™ BCA Protein Assay Kit (Thermo Fisher Scientific, USA). Then, 25 µg of protein extracts were separated by 13% sodium dodecyl sulfate-polyacrylamide gel electrophoresis (SDS-PAGE) and then transferred to a nitrocellulose membrane (Bio-Rad, USA), blocked with 5% whey protein in TBS-Tween buffer (0.12 M Tris-base, 1.5 M NaCl, 0.1% Tween 20). For primary antibody binding, the membrane was incubated with the appropriate primary antibodies (1:1000) (Santa Cruz Biotechnology Inc., Santa Cruz, CA, USA) overnight at 4 °C. Then, incubation was done with species-specific horseradish peroxidase (HRP)-labeled secondary antibodies (1:3000) (Thermo Scientific, USA) for an hour and subsequently washed in TBS-Tween (TBST). The blots were developed using a Chemiluminescent kit (Thermo Scientific, Rockford, lL, USA) to detect the target protein as per manufacturer’s protocol.

### 4.6. Caspase 8 Activity Assay

MCF-7 cells (1 × 10^6^) in logarithmic phage were treated with compound **A** (15, 25 and 50 μg/mL) for 24 h. The caspase activity was determined by using a Caspase 8 Assay Kit (Abcam, Cambridge, MA, USA) as per manufacturer’s protocol. Briefly, cells were washed with ice-cold PBS. Cells from untreated control and treated cells were lysed in ice-cold lysis buffer and kept on ice for 10 min. The lysate was centrifuged at 12,000× *g*, and the supernatant was transferred to a fresh tube. The protein concentrations of the supernatants were determined by a colorimetric assay using Pierce™ BCA Protein Assay Kit (Thermo Fisher Scientific, USA). A total of 200 μg of protein was used for further analysis. Then, 50 μL of 2X Reaction Buffer (containing 10 mM DTT) and 5 μL of the 1 mM IETD-AFC substrate were added to each sample and kept for incubation for 1.5 h. After, the fluorescence intensity was measured at a 400-nm excitation filter and 505-nm emission filter using a fluorescent plate reader.

### 4.7. Acute Toxicity Study of the Compound

Twelve clinically healthy Swiss albino mice, six weeks of age, were selected for this study. The mice were obtained from a commercial laboratory animal supplier and housed in polycarbonate cages kept at Laboratory Animal House of West Bengal University of Animal and Fishery Sciences, Kolkata, India. Sterile husk was used as bedding material for the mice, and the changing frequency was once a week. All mice were given access to water (RO and amp; UV treated) and standard feed (Nutri Lab, rodent feed, Vetcare Pvt. Ltd., Bangalore, India), both ad libitum. Mice were allowed to acclimatize for one week prior to the experiment. Then, the mice were randomly divided into two groups; each group was comprised of six mice. Group I was the control group which, was given only the vehicle (0.02% tween-20), single dose; Group II served as treatment group, which received compound **A** at 250 mg/kg, single dose, with a dose volume of 10 mL/kg. The experimental protocol was approved by the institutional animal ethical committee. Mice were kept on fast for 8 h prior to the administration of compound **A**. After administration, food was withheld for a further 3 h. Following administration, observations were made and recorded systematically, and a careful clinical examination was made once each day. Cage-side observations included changes in the skin and fur, eyes, respiratory, circulatory, autonomic and central nervous system, somatomotor activity, and behavior pattern. Particular attention was given to observe for tremors, convulsions, salivation, diarrhea, lethargy, sleep, coma, and death, if any. All mice were sacrificed humanely by cervical dislocation under mild anesthesia (Isoflurane, Baxter, San Juan, PR, USA). Necropsy was conducted on each animal for gross pathological changes. Biological samples (kidney, liver, spleen, intestine, and brain) were collected and preserved in a formalin solution (10%) for histopathology analysis. Serum samples were also collected for biochemical examination. The acute toxicity study was performed by a veterinary pathologist, and it was done blind-folded without any knowledge of the purpose of the current experiment.

### 4.8. Statistical Analysis

All data represent at least three independent experiments and are expressed as the mean ± standard deviation (SD). Statistical significance was determined by analysis of variance (ANOVA) using GraphPad Prism 5 software (http://www.graphpad.com). P-values of < 0.05 were considered statistically significant.

## Figures and Tables

**Figure 1 marinedrugs-17-00658-f001:**
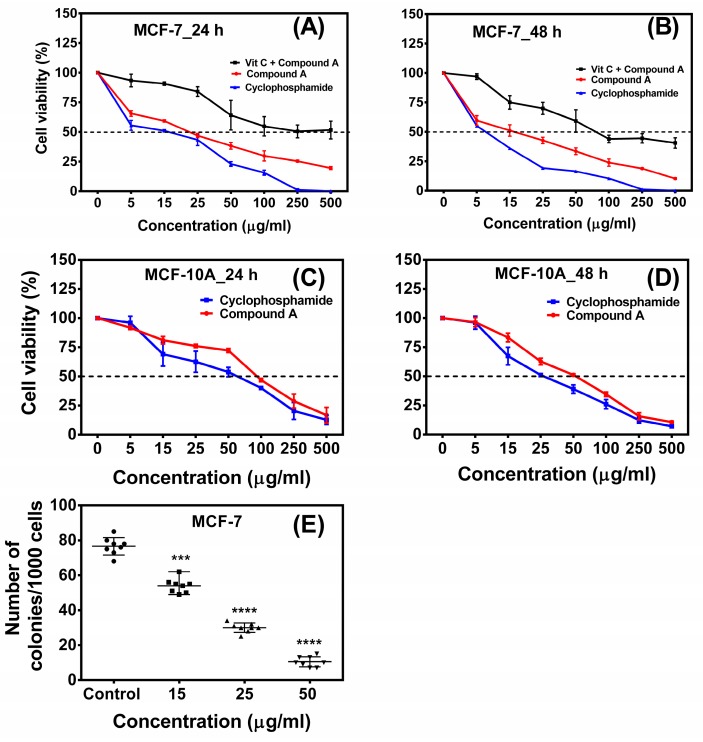
Cytotoxic effect of the quinazoline derivative (compound **A**) on breast carcinoma (MCF-7) and non-tumorigenic epithelial (MCF-10A) cell line. An MTT assay was done to evaluate the cytotoxic effect of the compound on MCF-7 and MCF-10A cell lines (**A**–**D**). Cyclophosphamide was used as a standard anti-cancer drug. (**A**) Effect of compound **A**, cyclophosphamide, and compound **A** + Vit C on viability of MCF-7 cells (24 h), (**B**) Effect of compound **A**, cyclophosphamide, and compound **A** + Vit C on viability of MCF-7 cells (48 h), (**C**) Effect of compound **A** and cyclophosphamide on viability of MCF-10A cells (24 h), (**D**) Effect of compound **A** and cyclophosphamide on viability of MCF-10A cells (48 h), (**E**) Effect of compound **A** on clonogenic potential of breast carcinoma cell line (MCF-7). Symbols (●,■,▲,▼) indicate number of colonies/1000 cells in control (0 µg/mL), 15 µg/mL, 25 µg/mL and 50 µg/mL compound **A** treated MCF-7 cells. Data are shown as mean ± SD. One-way analysis of variance (ANOVA) followed by Dunnett post-test was performed to find out significant difference among control and treatments; *** denotes *p* ≤ 0.001; **** denotes *p* ≤ 0.0001.

**Figure 2 marinedrugs-17-00658-f002:**
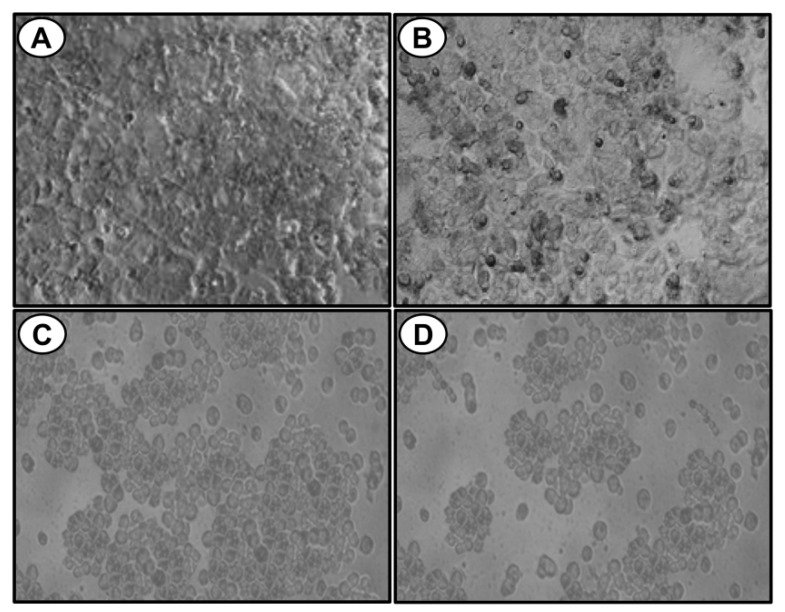
Bright field images of MCF-7 cells untreated (**A**) and treated with 15 µg/mL (**B**), 25 µg/mL (**C**), or 50 µg/mL (**D**) of compound **A**. A non-synchronized population of cells (10 × 10^3^) were plated in a six-well plate for 24 h before performing the cell treatment. Experiment was performed in triplicate.

**Figure 3 marinedrugs-17-00658-f003:**
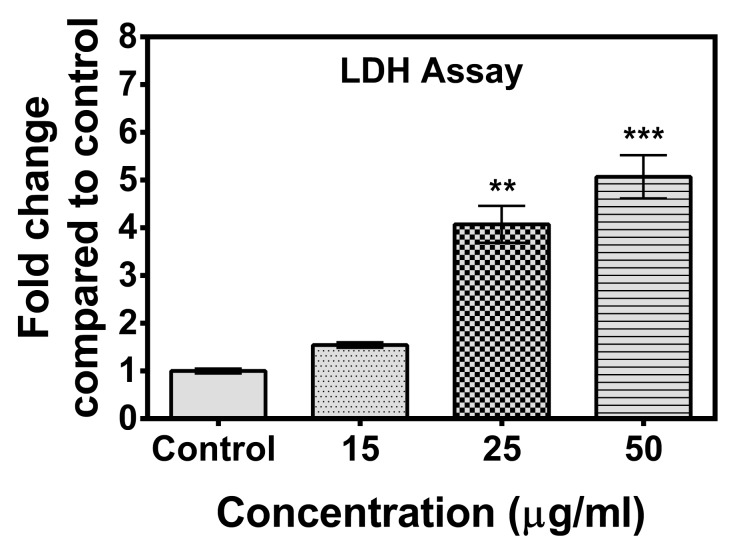
Lactate dehydrogenase (LDH) release assay. The assay revealed significant increase in LDH release following treatment of compound **A** (at concentrations 25 µg/mL and 50 µg/mL). The effect of the compound on the release of LDH was assessed after incubation for 48 h. Data are shown as mean ± SD. One-way analysis of variance (ANOVA) followed by Dunnett post-test was performed to find out significant difference among control and treatments. ** denotes *p* ≤ 0.01; *** denotes *p* ≤ 0.001.

**Figure 4 marinedrugs-17-00658-f004:**
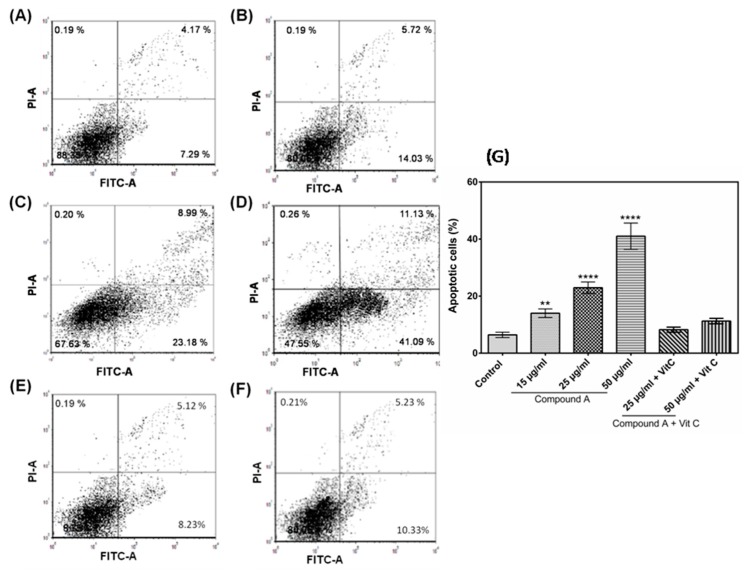
Apoptotic cells were stained by Annexin-V and detected by Flow Cytometry; concentration-dependent increase in percentage of apoptotic cells was found. (**A**) Control (0 µg/mL compound **A**), (**B**) 15 µg/mL compound **A**, (**C**) 25 µg/mL compound **A**, (**D**) 50 µg/mL compound **A**, (**E**) 25 µg/mL compound **A** + Vit C, (**F**) 50 µg/mL compound **A** + Vit C. (**G**) The histogram shows the mean Annexin V-positive MCF-7 cells (mean ± SD) of three experiments, One-way analysis of variance (ANOVA) followed by Dunnett post-test was performed to determine significant difference among control and treatments. ** denotes *p* ≤ 0.01; **** denotes *p* ≤ 0.0001.

**Figure 5 marinedrugs-17-00658-f005:**
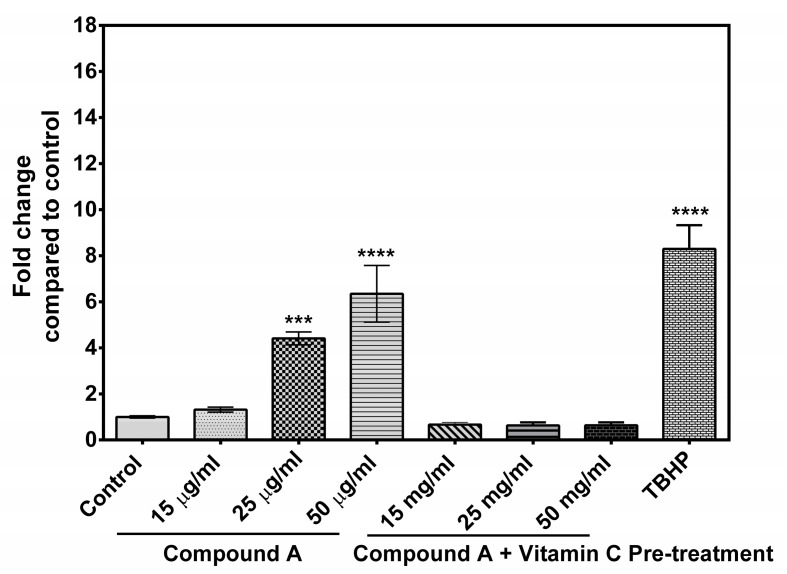
Effect of compound **A** on generation of reactive oxygen species (ROS). Tert-Butyl hydroperoxide (TBHP)-treated cells were taken as a positive control. The level of ROS was significantly elevated at concentration of 25 µg/mL. Data are shown as mean ± SD of three independent experiments. One-way analysis of variance (ANOVA) followed by Dunnett post-test was performed to determine significant difference among control and treatments. *** denotes *p* ≤ 0.001; **** denotes *p* ≤ 0.0001.

**Figure 6 marinedrugs-17-00658-f006:**
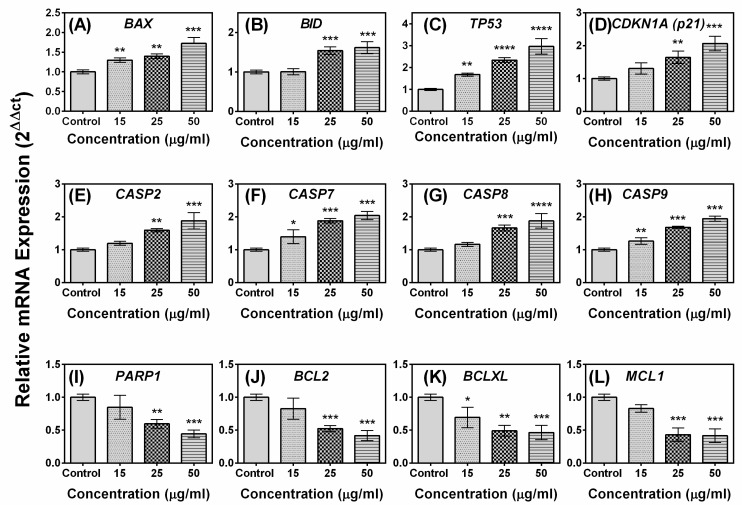
Effect of the compound on the expression of apoptosis related genes in MCF-7 cells. Cells were treated with compound **A** for 24 h and real-time RT-PCR was performed to analyze the expression of twelve genes involved in apoptosis pathway; (**A***) BAX*, (**B**) *BID*, (**C**) *TP53*, (**D**) *CDKN1A* (p21), (**E**) *CASP2*, (**F**) *CASP7*, (**G**) *CASP8*, (**H**) *CASP9*, (**I**) *PARP1*, (**J**) *BCL2*, (**K**) *BCLXL*, and (**L**) *MCL1*. The values and error bars represent average and standard deviations (SD) of three independent sets of experiments. One-way analysis of variance (ANOVA) followed by Dunnett post-test was performed to find out significant difference among control and treatments. * denotes *p*≤0.05; ** denotes *p* ≤ 0.01; *** denotes *p* ≤ 0.001; **** denotes *p* ≤ 0.0001.

**Figure 7 marinedrugs-17-00658-f007:**
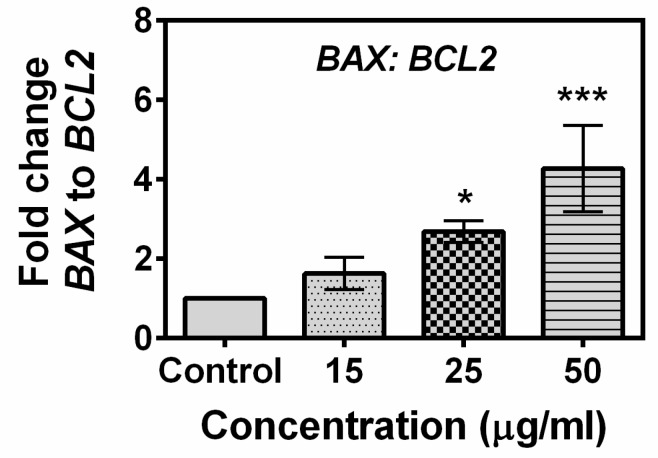
Effect of the compound on the ratio of *BAX* to *BCL2* gene expression in MCF-7 Cells. Cells were treated with compound **A** for 24 h. Real time RT-PCR was performed to analyze the expression of the genes, and their ratio was calculated. A significant increase in the ratio was observed when cells were treated with as low as 25 µg/mL of compound **A**. The values and error bars represent average and standard deviations (SD) of three independent sets of experiments. One-way analysis of variance (ANOVA) followed by Dunnett post-test was performed to find out significant difference among control and treatments. * denotes *p* ≤ 0.05, *** denotes *p* ≤ 0.001.

**Figure 8 marinedrugs-17-00658-f008:**
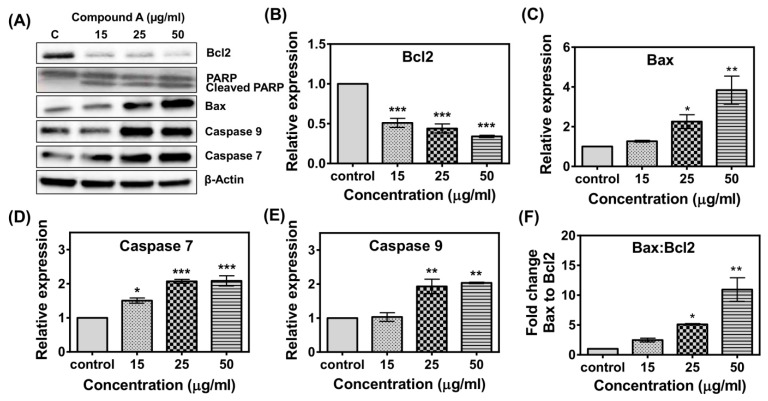
Western blot analysis of compound **A**–treated MCF-7 cells. Cells were treated with the compound for 24 h before being lysed and subjected to separation by sodium dodecyl sulfate (SDS) gel electrophoresis. Proteins were then transferred to a nitrocellulose membrane and probed with antibodies against Bcl-2, bax, caspase 7, caspase 9, and PARP. β-actin was used as a loading control. The band intensities of treated samples were normalized to β-actin. (**A**) The gel pic is a representative figure. The results indicated significant down regulation of Bcl2 (**B**) and up-regulation of Bax (**C**), Caspase 7 (**D**), and Caspase 9 (**E**). The ratio of Bax to Bcl2 was increased in treated cells (**F**). The values and error bars represent average and standard deviations (SD) of two independent sets of experiments. One-way analysis of variance (ANOVA) followed by Dunnett post-test was performed to find out significant difference among control and treatments. Quantification of bands (**B**–**E**) was done by Image J software [29]. * denotes *p* ≤ 0.05, ** denotes *p* ≤ 0.01, *** denotes *p* ≤ 0.001.

**Figure 9 marinedrugs-17-00658-f009:**
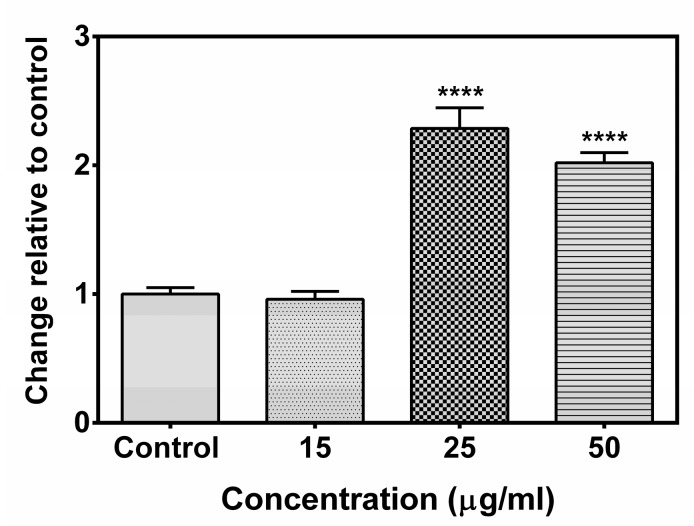
Effect of the compound on Caspase 8 activity. Cells were treated with compound **A** for 24 h. Caspase 8 activity was assessed by a colorimetric assay. One-way analysis of variance (ANOVA) followed by Dunnett post-test was performed to find out significant difference among control and treatments. **** denotes *p* ≤ 0.0001.

**Figure 10 marinedrugs-17-00658-f010:**
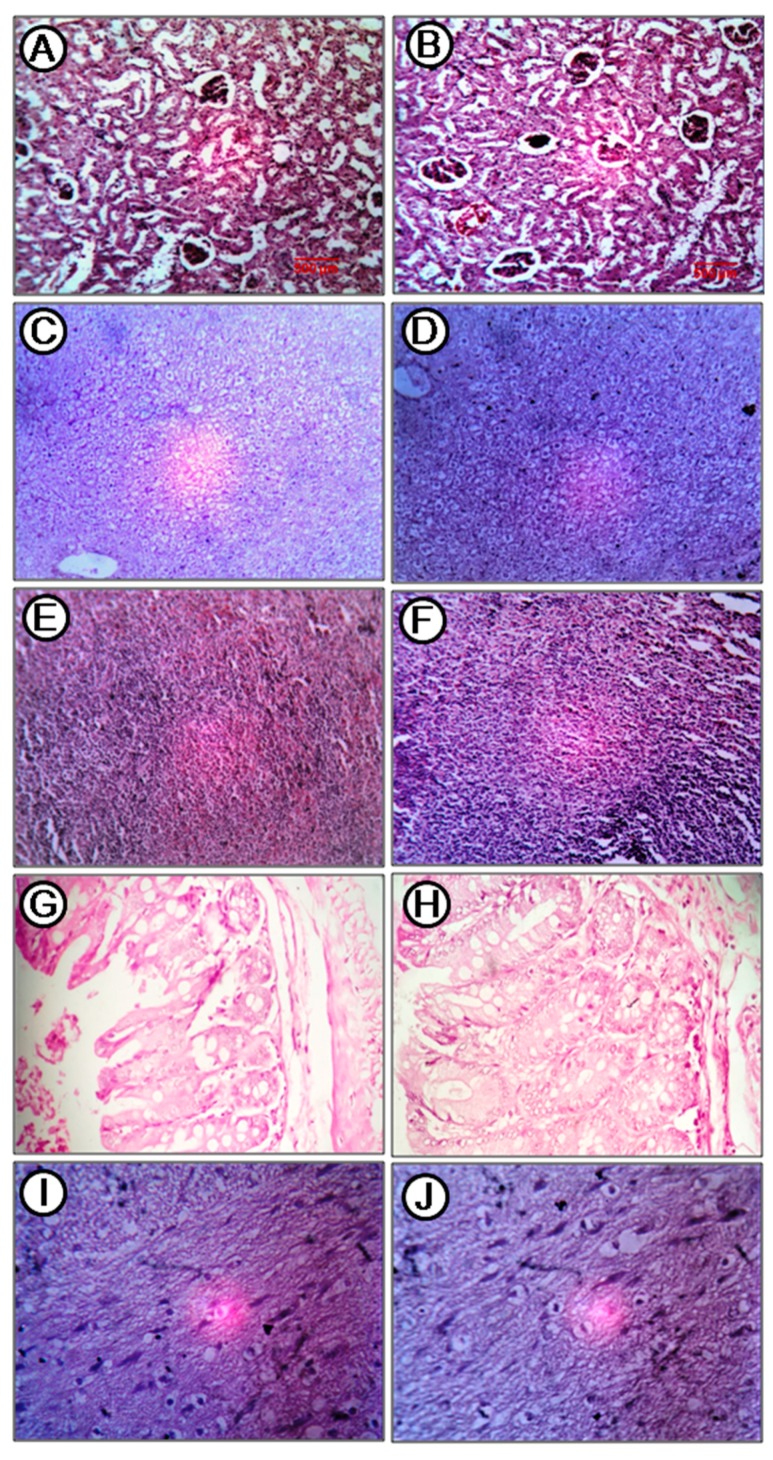
Histological sections in the acute toxicity test (Hematoxylin and eosin stain, H&E staining, 400×). (**A**) Kidney of control, (**B**) kidney of treatment, (**C**) liver of control, (**D**) liver of treatment, (**E**) spleen of control, (**F**) spleen of treatment, (**G**) intestine (cecum) of control, (**H**) intestine (cecum)of treatment, (**I**) brain of control, and (**J**) brain of treatment.

**Figure 11 marinedrugs-17-00658-f011:**
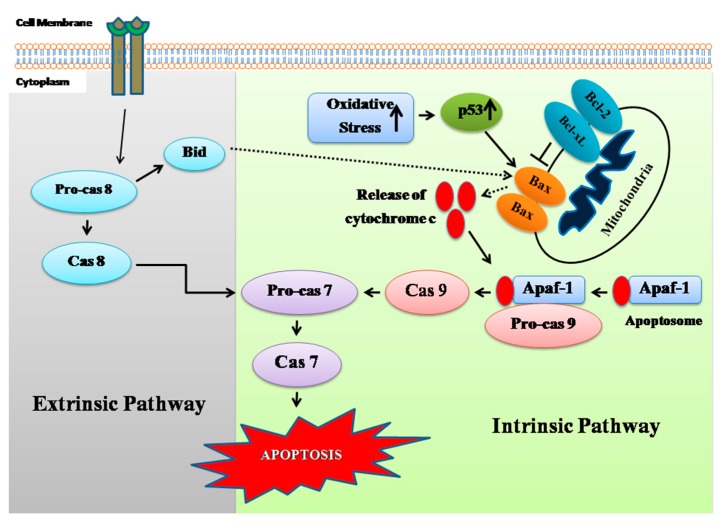
Schematic diagram of mechanism of cell death by the quinazoline derivative.

**Table 1 marinedrugs-17-00658-t001:** IC50 values of the quinazoline derivative (compound **A**) against MCF-7 and MCF-10A cell lines.

Compound	Cell Line	Cell Type	IC50 (µg/mL)	
			24 h	48 h
Compound **A**	MCF-7	Breast carcinoma cells	22.67 ± 1.53	13.04 ± 1.03
	MCF-10A	Non-tumorigenic epithelial cell	102.11 ± 1.89	51.25 ± 1.42
Compound **A** + Vit C	MCF-7	Breast carcinoma cells	249.67 ± 1.32	82.33 ± 1.39
Cyclophosphamide	MCF-7	Breast carcinoma cells	15.11 ± 1.16	8.11 ± 0.84
	MCF-10A	Non-tumorigenic epithelial cell	59.23 ± 1.68	26.22 ± 1.07

Data are presented as mean ± SD of three independent experiments.

**Table 2 marinedrugs-17-00658-t002:** Effect of the quinazoline derivative (compound **A**) on the liver function test.

Groups	Total Protein (g/L)	Albumin (g/L)	Globulin (g/L)	ALT (IU/L)	AST (IU/L)
Control (Vehicle)	53.0 ± 1.7	13.5 ± 0.65	53.7 ± 1.8	61.7 ± 6.4	264 ± 9.3
Treated (250 mg/kg)	51.6 ± 1.5	12.5 ± 0.72	54.4 ± 1.3	63.2 ± 4.5	262 ± 7.2

Data are presented as mean ± SD.

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
