# Peer review of "A Natural Quinazoline Derivative from Marine Sponge Hyrtios erectus Induces Apoptosis of Breast Cancer Cells via ROS Production and Intrinsic or Extrinsic Apoptosis Pathways"

_marinedrugs, 2019, doi:10.3390/md17120658_

Round 1

Reviewer 1 Report

I think the research is sound and authors did a fantastic job at addressing research questions.

In general, this paper is well-written and the authors addressed that natural quinazoline deivatives from this marine sponge induces apoptosis via ROS and intrinsic and extrinsic pathways with relevant assays. Concerning apoptosis, authors can also look at the effect of the compound on cleaved-caspase 3 and 7 since these proteins are indicative of apoptotic cell death. Also, authors would want to used other breast cancer cell lines to confirm the data obtained from the MCF-7 cell line.

Author Response

I think the research is sound and authors did a fantastic job at addressing research questions.

In general, this paper is well-written and the authors addressed that natural quinazoline deivatives from this marine sponge induces apoptosis via ROS and intrinsic and extrinsic pathways with relevant assays. Concerning apoptosis, authors can also look at the effect of the compound on cleaved-caspase 3 and 7 since these proteins are indicative of apoptotic cell death. Also, authors would want to used other breast cancer cell lines to confirm the data obtained from the MCF-7 cell line.

Response: We thank the reviewer for his/her enthusiasm for our study. As suggested, we have used another two mammary gland adenocarcinoma cell lines (MDA-MB-231 and MDA-MB-415) to confirm our data. MTT assay results showed that the inhibited the proliferation of the both the cell lines in a concentration dependent manner. The results are presented in Supplementary file (Supplementary Figure 1 and Supplementary table 1). The compound induced an up-regulation of pro-apoptotic genes BAX, BID, TR53, CDKN1A (p21), CASP2, CASP7, CASP8, CASP9 and PARP1 and down-regulation of anti-apoptotic genes BCL2, BCLXL, and MCL1 in MCF-7 cells. Protein level investigation indicated up-regulation of Bax, Caspase 7 and Caspase 9 and down-regulation of Bcl-2 following Compound A treatment. Moreover, the compound triggered cleavage of PARP. Taken together, it can be concluded that Compound A triggers apoptosis of MCF-7 cells.

Reviewer 2 Report

In this study, the authors showed the anti-proliferative effects, especially apoptotic pathway, of Hyrtios erectus-derived compound in breast cancer MCF-7. The in vitro experiments are thought to have been done carefully, but there are many problems to be solved.

As cited in Ref#63, the authors have already reported aa apoptosis pathway in MCF-7 induced by Hyrtios erectus-derived compound. In this their previous report, they reported that “Treatment of the sponge extract induced downregulation of antiapoptotic Bcl-2 protein and upregulation of Bax, caspase-3, caspase-9, and fragmented poly(ADP ribose)polymerase proteins in MCF-7 cells”. These findings are considered to be the main point of this present manuscript. The authors only describe this important previous report in the section of methods.

As the authors described in the discussion section, it is well-recognized that MCF-7 does not have caspase-3. Caspase-3 is also well-known to have important role in anticancer pathway in several types of cancer including breast cancer. The authors should clarify the reasons for using only MCF-7 and the concept of this study. Since apoptosis pathway in MCF-7 has already been elucidated by the authors, further studies using multiple cell lines are required to investigate this compound effects.

In this study, the authors examined acute toxicity of this compound in mice. The authors demonstrated the HE stanings in figure 9. These figures have never reached an acceptable level. Clear and high-magnification images are necessary to confirm the morphology of tissue and cells. I strongly recommend that the authors consult a veterinary or toxic pathologist on this toxicity study.

Author Response

In this study, the authors showed the anti-proliferative effects, especially apoptotic pathway, of Hyrtios erectus-derived compound in breast cancer MCF-7. The in vitro experiments are thought to have been done carefully, but there are many problems to be solved.

As cited in Ref#63, the authors have already reported aa apoptosis pathway in MCF-7 induced by Hyrtios erectus-derived compound. In this their previous report, they reported that “Treatment of the sponge extract induced downregulation of antiapoptotic Bcl-2 protein and upregulation of Bax, caspase-3, caspase-9, and fragmented poly(ADP ribose)polymerase proteins in MCF-7 cells”. These findings are considered to be the main point of this present manuscript. The authors only describe this important previous report in the section of methods.

Response: In the previous report, we studied the effect of the sponge extract on MCF-7 cell line and identified the active compounds present in the extract.  Five major alkaloid compounds were identified from the marine sponge extract. They were 5-hydroxy-3-(2-hydroxyethyl) indole, 1,6-dihydroxy-1,2,3,4-tetrahydro-β-carboline, hyrtiosin B, hyrtiosulawesine and 2-chloro-6-phenyl-8H-quinazolino[4,3-b]quinazolin-8-one. The effect of any individual compound was not studied. In the current study, we evaluated the anticancer potential of only one compound 2-chloro-6-phenyl-8H-quinazolino[4,3-b]quinazolin-8-one.

As the authors described in the discussion section, it is well-recognized that MCF-7 does not have caspase-3. Caspase-3 is also well-known to have important role in anticancer pathway in several types of cancer including breast cancer. The authors should clarify the reasons for using only MCF-7 and the concept of this study. Since apoptosis pathway in MCF-7 has already been elucidated by the authors, further studies using multiple cell lines are required to investigate this compound effects.

Response: As suggested, we have included another two mammary gland adenocarcinoma cell lines (MDA-MB-231 and MDA-MB-415). The antiproliferative activity of the compound on the two cell lines were studied by MTT assay and it was found that the inhibited the proliferation of the both the cell lines in a concentration dependent manner (Supplementary Figure 1 A-D). The apoptotic potential of the compound against the two cell lines was studied by Flow cytometry assay (Supplementary Figure 2) and it was found that the compound induced apoptosis of the two cell lines in a concentration dependent manner. In summary, it may be concluded that the compound has apoptotic potential against breast cancer cells.

In this study, the authors examined acute toxicity of this compound in mice. The authors demonstrated the HE stanings in figure 9. These figures have never reached an acceptable level. Clear and high-magnification images are necessary to confirm the morphology of tissue and cells. I strongly recommend that the authors consult a veterinary or toxic pathologist on this toxicity study.

Response: The acute toxicity study was done by a veterinary pathologies and it was done blind folded without any knowledge of the purpose of the current experiment. No significant variation in behaviour or histological changes between treatment group and control group was reported.

Reviewer 3 Report

In the present manuscript Kumar De et al describe anti-proliferative effect of natural quinazoline derivative (2-chloro-6-phenyl-8H-quinazolino[4,3-b]quinazolin-8-one, compound A) isolated from marine sponge Hyrtios erectus on human breast carcinoma cell line (MCF-7). They report that compound A induce apoptosis and increase pro-apoptotic gene/protein and decrease anti-apoptotic gene/protein on MCF-7. Moreover, they describe that compound A induce oxidative stress and that anti-proliferative effect of compound A is ROS mediated.  In the end they performed an acute toxicity test with compound A in an mouse model.

The manuscript could be interesting especially for the minor anti-proliferative effect on non-tumorigenic epithelial cell line (MCF-10A) but needs major improvements.

Some major points:

In Fig 1 and Table 1 cyclophosphamide is present but it is missing in the text and in figure legend. The authors could add a comment in the text. LHD assay reflects disruption of plasma membrane and not apoptosis. LDH is rapidly released into the cell culture supernatant when the plasma membrane is damaged, a key feature of cells undergoing apoptosis, necrosis, and other forms of cellular damage. (Kumar P. Analysis of Cell Viability by the Lactate Dehydrogenase Assay. Cold Spring Harb Protoc. 2018 Jun 1;2018(6)). The authors could correct the text. Propidium iodide (PI) uptake reflects disruption of plasma membrane as LHD assay. Moreover, the percentage of PI-positive cells after 24h (Fig 3 B, C, D,) is lower than that after 6h (Fig 3 F) of compound A I would suggest removing Fig 3F (and paragraph 2.3) because it is redundant and contradictory. In title, abstract and, conclusion the authors write that apoptosis is mediate by ROS production but they perform only MTT cytotoxicity assay on MCF-7 treated with compound A and Vit C. The authors could perform an Annexin-V/PI assay on MCF-7 treated with compound A in presence of Vit C to evaluate the role of ROS in apoptosis. Since the different anti-proliferative effect of compound A between MCF-7 and MCF-10A is interesting, the authors should investigate apoptosis and ROS production on MCF-10A treated with compound A and compare the results with those obtained on MCF-7. The Fig 10 could be improved on the basis of experimental data; furthermore, p53 increases after treatment with compound A. The authors could improve figure legends and Materials and Methods, missing primer sequences, antibody specifications, wavelengths, reagent amounts, time of treatments….

Some minor points:

MCF-7 is a breast carcinoma cell line and not primary breast cancer cells. MCF10A cell line is a non-tumorigenic epithelial cell line and not normal mammary epithelial cells because cell origin is fibrocystic disease (Ying Qu et al. Evaluation of MCF10A as a Reliable Model for Normal Human Mammary Epithelial Cells. PLoS One. 2015 Jul 6;10(7):e0131285.). Information present in line 29 is related to cancer in general and not to breast I would suggest removing line 270-71 because it is redundant. I would suggest to specify in text compound A and not only compound. In axis of fig 5 "-" is missing. Fig 7 B-E is a quantification of the intensity of the bands of fig 7A, the authors should indicate how it was obtained There are some errors in the English form.

Author Response

In the present manuscript Kumar De et al describe anti-proliferative effect of natural quinazoline derivative (2-chloro-6-phenyl-8H-quinazolino[4,3-b]quinazolin-8-one, compound A) isolated from marine sponge Hyrtios erectus on human breast carcinoma cell line (MCF-7). They report that compound A induce apoptosis and increase pro-apoptotic gene/protein and decrease anti-apoptotic gene/protein on MCF-7. Moreover, they describe that compound A induce oxidative stress and that anti-proliferative effect of compound A is ROS mediated.  In the end they performed an acute toxicity test with compound A in an mouse model.

The manuscript could be interesting especially for the minor anti-proliferative effect on non-tumorigenic epithelial cell line (MCF-10A) but needs major improvements.

Response: We thank the reviewer for his/her enthusiasm for our study.

Some major points:

In Fig 1 and Table 1 cyclophosphamide is present but it is missing in the text and in figure legend. The authors could add a comment in the text.

Response: Necessary modification has been done.

LHD assay reflects disruption of plasma membrane and not apoptosis. LDH is rapidly released into the cell culture supernatant when the plasma membrane is damaged, a key feature of cells undergoing apoptosis, necrosis, and other forms of cellular damage. (Kumar P. Analysis of Cell Viability by the Lactate Dehydrogenase Assay. Cold Spring Harb Protoc. 2018 Jun 1;2018(6)). The authors could correct the text.

Response: Necessary corrections have been made and incorporated in the revised manuscript.

Propidium iodide (PI) uptake reflects disruption of plasma membrane as LHD assay. Moreover, the percentage of PI-positive cells after 24h (Fig 3 B, C, D,) is lower than that after 6h (Fig 3 F) of compound A I would suggest removing Fig 3F (and paragraph 2.3) because it is redundant and contradictory.

Response: Fig. 3F and paragraph 2.3 were deleted in the revised manuscript.

In title, abstract and, conclusion the authors write that apoptosis is mediate by ROS production but they perform only MTT cytotoxicity assay on MCF-7 treated with compound A and Vit C. The authors could perform an Annexin-V/PI assay on MCF-7 treated with compound A in presence of Vit C to evaluate the role of ROS in apoptosis.

Response: As suggested it was done and included in revised manuscript (Fig. 4 E-G) and paragraph 2.3.

 Since the different anti-proliferative effect of compound A between MCF-7 and MCF-10A is interesting, the authors should investigate apoptosis and ROS production on MCF-10A treated with compound A and compare the results with those obtained on MCF-7.

Response: Effect of the compound on apoptosis and generation of ROS in MCF-10A cells was evaluated and the results are included in paragraph 2.3, 2.4 and Supplementary section (Supplementary Figure 3 and  Supplementary Figure 4).

The Fig 10 could be improved on the basis of experimental data; furthermore, p53 increases after treatment with compound A. The authors could improve figure legends and Materials and Methods, missing primer sequences, antibody specifications, wavelengths, reagent amounts, time of treatments….

Response: Necessary corrections have been made and incorporated in the revised manuscript.

Some minor points:

MCF-7 is a breast carcinoma cell line and not primary breast cancer cells. MCF10A cell line is a non-tumorigenic epithelial cell line and not normal mammary epithelial cells because cell origin is fibrocystic disease (Ying Qu et al. Evaluation of MCF10A as a Reliable Model for Normal Human Mammary Epithelial Cells. PLoS One. 2015 Jul 6;10(7):e0131285.).

Response: Modifications have been made.

 Information present in line 29 is related to cancer in general and not to breast I would suggest removing line 270-71 because it is redundant. I would suggest to specify in text compound A and not only compound.

Response: Modifications have been made.

 In axis of fig 5 "-" is missing. Fig 7 B-E is a quantification of the intensity of the bands of fig 7A, the authors should indicate how it was obtained There are some errors in the English form. 

Response: Modifications have been made.

Reviewer 4 Report

The manuscript titled “A natural quinazoline derivative from marine sponge Hyrtios erectus induces apoptosis of breast cancer cells via ROS production and intrinsic and extrinsic apoptosis pathways” by De and colleagues present data that indicates the isolated quinazoline compound has cytotoxic effects on the MCF7 breast cancer cell line.  The data supports a mechanism of quinazoline-induced ROS production that leads to caspase-dependent apoptosis.

While the data presented is intriguing, the logical flow of the paper is completely missing.  The data discussed in the results sections jumps from figure to figure and presents the last panel of a figure before the first.  It is confusing and distracts from a decent story.  For example, in section 2.1 for figure 1, the results jump from MTT assays to ROS and the results mention apoptosis without initially producing evidence that ROS and apoptosis are actually involved in the quinazoline-induced reduction in mitochondrial processing of MTT.  It makes the entire manuscript confusing to read.

The statement ending in line 95 that the quinazoline treatment has “cytotoxic activity against primary breast cancer cells” is misleading.  MCF7 cells are not primary cancer cells; MCF7 cells are a cell line.  This type of misleading wording happens throughout the manuscript.

The statement throughout the manuscript that MCF10A cells are “normal mammary epithelial cells” is very misleading.  While not a breast cancer cell line, MCF 10A cells are not a normal mammary epithelial cell as stated, but are a non-transformed mammary epithelial cell line.  MCF 10A are an immortalized mammary epithelial cell line that exhibit many different growth properties than a true normal mammary epithelial cell.  This needs to be corrected throughout the manuscript.

The data do not support the conclusion that there is a time-dependent effect of quinazoline treatment.  Two time points plotted on independent graphs do not demonstrate time-dependent effects.

The conclusion that both the intrinsic and extrinsic apoptotic pathways are activated by the quinazoline treatment of MCF7 cells is not substantiated by the data.  Only activation of caspase 8 (the extrinsic pathway) is shown.  Upregulation of the caspase 7 gene as shown by qPCR and western blot analyses does not show that the intrinsic pathway is activated.  Caspases are expressed as pro-genes that must be cleaved to be activated and this is not shown.  It is also not clear why both arms of the apoptotic pathway would need to be activated in response to the quinazoline treatment.  Is this an indication that the cells are just necrotic rather than apoptotic?

For the caspase 8 activity assay shown in figure 8, where is the positive control for known induced caspase 8 activity in MCF7 cells?

Is there any information on how quinazoline induces ROS production?  This is a “black box” in the presented model and no information is offered to help explain how ROS are induced to start this cytotoxic process.

The statement that p53-induced p21 leads to apoptosis of cancer cells (line 280) is misleading.  P53-induced p21 can also lead to cellular senescence.  The biological result of this pathway is context dependent.  There is no data presented demonstrating that quinazoline-induced p53 leads to p21 which leads to apoptosis.  At best these are true, true, and unrelated data.

Minor Points –

In the model shown in figure 10, The dark blue circle (Bid?) with the arrow pointing to it from Caspase 8 on the upper left side is impossible to read.

Author Response

The manuscript titled “A natural quinazoline derivative from marine sponge Hyrtios erectus induces apoptosis of breast cancer cells via ROS production and intrinsic and extrinsic apoptosis pathways” by De and colleagues present data that indicates the isolated quinazoline compound has cytotoxic effects on the MCF7 breast cancer cell line.  The data supports a mechanism of quinazoline-induced ROS production that leads to caspase-dependent apoptosis.

Response: We thank the reviewer for his/her enthusiasm for our study.

While the data presented is intriguing, the logical flow of the paper is completely missing.  The data discussed in the results sections jumps from figure to figure and presents the last panel of a figure before the first.  It is confusing and distracts from a decent story.  For example, in section 2.1 for figure 1, the results jump from MTT assays to ROS and the results mention apoptosis without initially producing evidence that ROS and apoptosis are actually involved in the quinazoline-induced reduction in mitochondrial processing of MTT.  It makes the entire manuscript confusing to read.

 Response: Necessary modifications have been made.

The statement ending in line 95 that the quinazoline treatment has “cytotoxic activity against primary breast cancer cells” is misleading.  MCF7 cells are not primary cancer cells; MCF7 cells are a cell line.  This type of misleading wording happens throughout the manuscript.

Response: Necessary modifications have been made.

 The statement throughout the manuscript that MCF10A cells are “normal mammary epithelial cells” is very misleading.  While not a breast cancer cell line, MCF 10A cells are not a normal mammary epithelial cell as stated, but are a non-transformed mammary epithelial cell line.  MCF 10A are an immortalized mammary epithelial cell line that exhibit many different growth properties than a true normal mammary epithelial cell.  This needs to be corrected throughout the manuscript.

 Response: Necessary modifications have been made.

The data do not support the conclusion that there is a time-dependent effect of quinazoline treatment.  Two time points plotted on independent graphs do not demonstrate time-dependent effects.

Response: Necessary modifications have been made.

The conclusion that both the intrinsic and extrinsic apoptotic pathways are activated by the quinazoline treatment of MCF7 cells is not substantiated by the data.  Only activation of caspase 8 (the extrinsic pathway) is shown.  Upregulation of the caspase 7 gene as shown by qPCR and western blot analyses does not show that the intrinsic pathway is activated.  Caspases are expressed as pro-genes that must be cleaved to be activated and this is not shown.  It is also not clear why both arms of the apoptotic pathway would need to be activated in response to the quinazoline treatment.  Is this an indication that the cells are just necrotic rather than apoptotic?

Response: In the present study, it was found that the compound induced increased gene expression of caspase 2, 7, 8 and 9 (Figure 6). In western blot analysis, increased expression of Caspase 7 and 9 was detected in the compound treated breast cancer cells (Figure 8). Caspase 7, Caspase 9 are involved in intrinsic pathway of apoptosis, whereas Caspase 8 is involved in extrinsic pathway of apoptosis. It was also found that the compound induced Caspase 8 activity as detected in Caspase 8 assay (Figure 9). Therefore, it is highly plausible that the compound activate either intrinsic or extrinsic or both the pathways of apoptosis. Induction of apoptosis following Compound A treatment was confirmed by Flow cytometry assay (Figure 4).

 For the caspase 8 activity assay shown in figure 8, where is the positive control for known induced caspase 8 activity in MCF7 cells?

Response: The change in caspases 8 activity was determined by comparing these results with the level of the uninduced control.

Is there any information on how quinazoline induces ROS production?  This is a “black box” in the presented model and no information is offered to help explain how ROS are induced to start this cytotoxic process.

Response: The mechanism of ROS generation by the quinazoline derivative merits further study.

The statement that p53-induced p21 leads to apoptosis of cancer cells (line 280) is misleading.  P53-induced p21 can also lead to cellular senescence.  The biological result of this pathway is context dependent.  There is no data presented demonstrating that quinazoline-induced p53 leads to p21 which leads to apoptosis.  At best these are true, true, and unrelated data.

Response: The portion has been revised.

Minor Points –

In the model shown in figure 10, The dark blue circle (Bid?) with the arrow pointing to it from Caspase 8 on the upper left side is impossible to read.

Response: Necessary modification has been made.

Round 2

Reviewer 2 Report

Although the manuscript was considerably improved, unfortunately, pathology remained untouched. The readers will not be able to convince the authors' argument regarding the histology. 

Author Response

Although the manuscript was considerably improved, unfortunately, pathology remained untouched. The readers will not be able to convince the authors' argument regarding the histology.

Response: The histopathology figures have been modified as suggested by the reviewer.

Reviewer 3 Report

The article has undergone significant improvements. Some corrections are needed:

Fig3: the autors wrote: “ The assay revealed significant cytotoxicity of the compound on MCF-7 cells at 25 μg/ml and pretreatment with Vit C (0.1 mM) increase the viability of MCF-7 cells” but in figure is not present the effect of compound A and pretreatment. Fig4: the statistical analysis and SD are missing on new experimental conditions. Suppl Fig2. there is an error in the text (MDA-MB-231 described twice).

Author Response

Fig3: the autors wrote: “ The assay revealed significant cytotoxicity of the compound on MCF-7 cells at 25 μg/ml and pretreatment with Vit C (0.1 mM) increase the viability of MCF-7 cells” but in figure is not present the effect of compound A and pretreatment. Fig4: the statistical analysis and SD are missing on new experimental conditions. Suppl Fig2. there is an error in the text (MDA-MB-231 described twice).

Response: Necessary modification has been made in Fig. 3 legend. Fig. 4: Statistical analysis including SD has been done and included. No significant difference was observed between control and treatment (25 µg/ml + VitC) or control and 50 µg/ml + VitC. In Suppl Fig 2, the error in text has been rectified.

Reviewer 4 Report

The authors have addressed  the concerns raised in the original review.

Author Response

The authors have addressed  the concerns raised in the original review.

Response: The authors would like to thank the reviewer for his/her comments to improve the manuscript.